# Validation of the actuator disc approach using small scale model wind turbines

Nikolaos Simisiroglou[1,2], Simon-Philippe Breton[2], and Stefan Ivanell[2]

[1]WindSim AS, Fjordgaten 15, N-3125 Tønsberg , Norway
[2]Uppsala University, Wind Energy Section, Campus Gotland, 621 67 Visby, Sweden

*Correspondence to:* Nikolaos Simisiroglou (nikolaos.simisiroglou@windsim.com)

**Abstract.** The aim of the present study is the validation of the implementation of an actuator disc model in the CFD code PHOENICS. The flow behaviour for three wind turbine cases is investigated numerically and compared to wind tunnel measurements: (A) the flow around a single model wind turbine, (B) the wake interaction between two in–line model wind turbines for a uniform inflow of low turbulence intensity and (C) the wake interaction between two in–line model wind turbines at different separation distances in a uniform or sheared inflow of high turbulence intensity. This is done by using Reynolds averaged Navier–Stokes (RANS) and an actuator disc (ACD) technique in the computational fluid dynamics code PHOENICS. The computations are conducted for the design condition of the rotors using four different turbulence closure models and five different thrust distributions. The computed axial velocity field as well as the turbulent kinetic energy are compared with hot wire anemometry (HWA) measurements. For the in–line two model wind turbine cases, the thrust coefficient is also computed and compared with measurements. The results show that for different inflow conditions and wind turbine spacings the proposed method is able to predict the overall behaviour of the flow with low computational effort. When using the $k - \varepsilon$ and $KL\, k - \varepsilon$ turbulence models the results are generally in closer agreement with the measurements.

## 1 Introduction

The study of wake properties is important for assessing the optimal layout of modern wind farms. Wind turbine wake development may be studied using field experiments, small scale wind tunnel measurements or numerical simulations with computational fluid dynamics (CFD). There are several advantages of CFD over field experiments and small scale wind tunnel measurements e.g. no violation of similarity requirements, control over inflow conditions and whole flow field data of relevant parameters. However as CFD results are sensitive to the experience and knowledge of the user of the CFD code and to the numerous computational parameters and assumptions involved, it is imperative to perform validation studies. Previous work on validating CFD wake models using a wind turbine tested in wind tunnels have been presented by Simms et al. (2001) and by Schepers et al. (2012). These studies demonstrated that there was a significant deviation between the various prediction tools and the wind tunnel measurements. Similar results for a small scale model wind turbine are reported by Krogstad and Eriksen (2013) and by Pierella et al. (2014) indicating the importance of validating existing wind turbine modelling tools and methodologies.

Advanced methods of wake modelling with CFD may be done by using large eddy simulation (LES) techniques in which the wind turbine forces may either be prescribed with an actuator line method (ACL) or an ACD method. Work along these lines has been performed by numerous researchers such as Breton et al. (2014); Nilsson et al. (2015); Churchfield et al. (2012); Andersen et al. (2015); Calaf et al. (2010); Ivanell et al. (2007). Although LES provides high fidelity results comparable to field measurements, the computational requirements of the method (Churchfield et al. (2012); Laan et al. (2015b)) is still too expensive and therefore not yet suitable for engineering practices of whole wind farm wake computations. A less computationally expensive alternative to LES are Reynolds averaged Navier-Stokes (RANS) simulations. RANS simulations have been used with the ACD method to simulate wind turbine wakes by numerous researchers, e.g. Laan et al. (2015a); Prospathopoulos et al. (2011); El Kasmi and Masson (2008).

The aim of the present study is the validation of the implementation of an actuator disc model in the CFD code PHOENICS Spalding (1981). In order to do so, computational and experimental results are compared for three cases. Case A consists of a single wind turbine in a low turbulence intensity environment with a uniform wind inflow. Case B is composed of two wind turbines positioned in-line in the same low turbulence intensity environment with a uniform wind inflow. Case C again uses two wind turbines positioned in-line, but in this case multiple inflow conditions are studied for different spacings of the wind turbines. This is done to investigate the influence of the inlet conditions on the wind turbines thrust.

As this method is intended to be used for industrial purposes, it therefore needs to provide accurate and reliable results with low computational effort. The simulations are performed according to the "Blind test 1", "Blind test 2" and "Blind test 4" invitation workshops organised by NOWITECH and NORCOWE (Krogstad et al. (2011); Pierella et al. (2012); Sætran and Bartl (2015)). The goal of these three workshops is to serve as an ideal test case for CFD tools by providing detailed measurements of the thrust coefficient and the wake properties behind the rotor both in terms of mean flow and turbulence kinetic energy within a controlled wind tunnel environment. Note that this work is an extension of the proceeding Simisiroglou et al. (2016).

The paper unfolds as follows; section 2 presents the experimental set-up of the workshops in which the three test cases are outlined, this is followed by a description of the numerical method and of the computational settings used to perform the simulations. The results from the numerical simulations are introduced and discussed in section 3. Lastly, in section 4 the main conclusions of this study are presented.

## 2 Methods

### 2.1 Experimental set–up

The experiments are performed in the large closed–return wind tunnel facility at the Norwegian University of Science and Technology (NTNU). The test section for all three cases has the width ($W$) and length ($L$) dimensions of $W \times L = 2.710$ m $\times$ 11.150 m, Fig. 1. To maintain a zero pressure gradient and maintain a constant velocity along the stream–wise direction the height $H$ of the wind tunnel increases from 1.801 m at the inlet to 1.851 m at the outlet. Velocity measurements are performed using both hot wire anemometry (HWA) and Laser–Doppler Anemometry for verification purposes. The tip Reynolds number

for these three cases is approximately $Re_{c,tip} \approx 10^5$ for the upstream wind turbine. This tip Reynolds number is based on the velocity of the tip and the chord length at tip. For full scale experiments a typical tip Reynolds number is in the order of $10^6$. The air density $\rho$ is equal to 1.2 kg m$^{-3}$. In all cases the results are only considered where the wind turbines are operating at their design condition i.e. tip speed ratio of six (TSR=6).

In case A, the three–bladed wind turbine is positioned at a distance of 3.660 m from the inlet. The model wind turbine has a tower that consists of four cylinders of different radii. The hub height is $H_{hub} = 0.817$ m and the rotor radius is $R = 0.447$ m. The rotor blades are designed to produce a constant pressure drop across the rotor, which resembles a uniformly distributed thrust, when operating at their design condition (Krogstad and Eriksen (2013)). The airfoil used is the NREL S826 and to increase the Reynolds number of the blades, a chord length of approximately three times longer than normal was used. The

blades have a circular shape close to the nacelle primarily to allow them to be attached to the hub. The transition from the airfoil section of the blade to the circular section is abrupt. An asynchronous generator of 0.37 kW is located under the tunnel floor and is connected to the wind turbine rotor by a belt located behind the tower. The total blockage effect, defined herein as the fraction of the total tower and rotor swept area to the wind tunnel cross section, is approximately 13%. As a result the flow will be impacted by the walls and this interference will lead to artificial speed up effects. The stream–wise inlet

velocity is $U_{ref,A} = 10$ m s$^{-1}$ and the stream–wise turbulence intensity at the turbine position is $I_{u,A} = 0.3\%$. A thin boundary exists near the wall of the wind tunnel. This boundary layer has been measured in an empty tunnel using pitot tubes for four distances downstream of the wind tunnel inlet i.e. 1.80 m, 4.50 m, 6.30 m and 8.10 m. Further information on the details of the experimental investigations are reported by Krogstad and Adaramola (2012) and Krogstad and Lund (2012).

For case B the stream–wise inlet velocity and turbulence intensity are similar to case A. Here two in-line wind turbines

horizontally centred in the wind tunnel are investigated, where the downwind wind turbine is the same one as used in case A. The upstream wind turbine hereafter is always referred to as $T_1$ while the downstream as $T_2$. Both wind turbines rotate in a counter–clockwise direction as seen from the inlet and are three bladed with the same blade geometry and airfoil i.e. the NREL S826 airfoil. As the nacelle diameter of $T_1$ is somewhat larger than $T_2$, the turbines have slightly different rotor diameters. The rotor radius of the upstream wind turbine $T_1$, is $R_{T_1} = 0.472$ m and the radius of the downstream wind turbine $T_2$, is

$R_{T_2} = R = 0.447$ m. For $T_1$ as opposed to $T_2$ the belt connected to the 0.37 kW asynchronous generator is located within the tower. To calculate the thrust force of the turbines, they are mounted on a six–component force balance. Further information on the details of the experimental investigations are reported by Pierella et al. (2012) and Pierella et al. (2014).

Case C is divided into two sub–cases C1 and C2. For both sub–cases the same wind turbines as in case B are used with a hub height of 0.827 m instead of 0.817 m. The distinction between sub–cases is done because the wind turbines are exposed

to different inflow conditions in terms of the wind velocity profile and turbulence intensity as seen in Table 1. For both sub–cases the upstream wind turbine is positioned at $4R$ from the inlet. Looking at each case individually, case C1 has a uniform inflow velocity of $U_{ref,C} = 11.5$ m s$^{-1}$ measured at the inlet and a turbulence intensity of 10% measured at the first wind turbine position. The turbulence in the wind tunnel is created by a bi–planar grid built from wooden bars installed at the inlet. To estimate the effect of unintended stream–wise velocity gradients, horizontal stream–wise velocity values at four positions

downstream of the inlet are measured in an empty domain. For case C1 the thrust values along with the velocity and turbulence

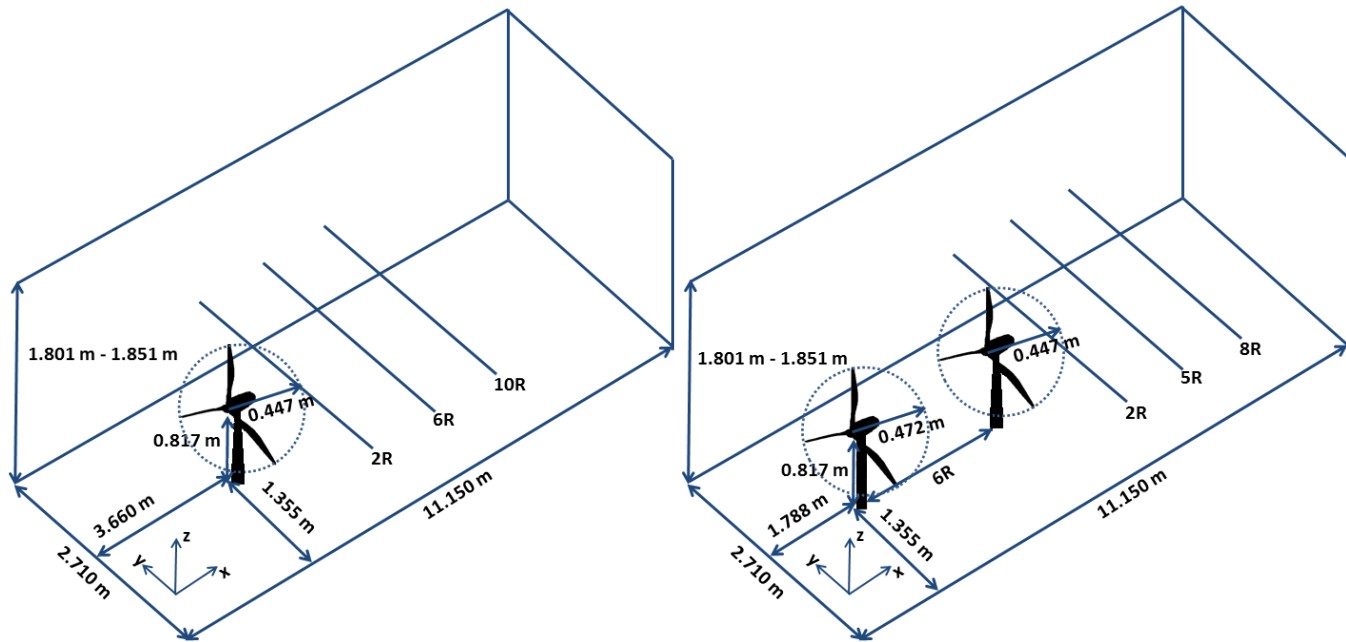

**Figure 1.** Illustration of wind tunnel layout for (a) one model wind turbine set–up (case A), (b) two in-line model wind turbine set–up (case B). The three downstream positions $x/R = 2, 6, 10$ and $x/R = 2, 5, 8$ are where the measurements are extracted, radius $R = 0.447$ m.

kinetic energy at three distances downstream of $T_1$ are measured. Regarding case C2, a sheared inflow is considered with a turbulence intensity of 10.1% at the hub height position of $T_1$. The inlet velocity as a function of height $z$ is expressed by the power law used for atmospheric flows which is given as

$$U(z) = U(z_{ref}) \left( \frac{z}{z_{ref}} \right)^{a},$$
(1)

where $a$ is the shear exponent equal to 0.11, the reference height is $z_{ref} = 0.827$ m and reference velocity is $U(z_{ref}) = U_{ref,C} = 11.5$ m s$^{-1}$ . For C2, similar to case C1, empty domain measurements are conducted at the same four positions from the inlet for the horizontal stream–wise velocities and turbulence intensity. The position of the second wind turbine is fixed to $10.36R$ and wake measurements for stream–wise velocity and turbulent kinetic energy are taken at a downstream distance of $T_1$ equal to $5.54R$.

## 2.2 Numerical method

The simulations are performed with the commercial CFD code PHOENICS in which the Reynolds Averaged Navier-Stokes equations (RANS) are solved using four different turbulence models. The turbulence models are (1) the standard $k - \varepsilon$ Launder and Spalding (1974), (2) the $RNG$ $k - \varepsilon$ Yakhot and Smith (1992), (3) the $KL$ $k - \varepsilon$ Kato (1993) and (4) Wilcox's $k - \omega$ turbulence model Wilcox (1988). The flow variables are stored in a uniform fully structured staggered grid and the Cartesian coordinate system is used. The SIMPLEST algorithm Spalding (1980) is used to solve the RANS equations and the hybrid

**Table 1.** Overview of cases.

| Case | Separation distance $x/R$ | Wind profile | Turbulence intensity | Hub height[m] | Measurement position $x/R$ |
|------|---------------------------|--------------|----------------------|---------------|----------------------------|
| A | – | Uniform 10 m s$^{-1}$ | 0.3% | 0.817 | 2, 6, 10 |
| B | 6 | Uniform 10 m s$^{-1}$ | 0.3% | 0.817 | 2, 5, 8 |
| $C_1$ | 18.00 | Uniform 11.5 m s$^{-1}$ | 10% | 0.827 | 5.54, 10.36, 17.00 |
| $C_2$ | 10.36 | Sheared 11.5 m s$^{-1}$ at hub | 10.1% | 0.827 | 5.54 |

differencing scheme Spalding (1972) is used to discretize the convective terms. The diffusion terms are discretized using the central differencing scheme. In the computations, the wind tunnel conditions are replicated accordingly for each case and a zero static pressure is applied at the outlet plane. The lateral, top and bottom faces of the domain are set to be impermeable and a wall function method according to Launder and Spalding (1974) is employed to introduce the effects of the wind tunnel walls

into the numerical simulation. This particular method is preferred for its advantages in terms of low computational requirements and storage needs. The wall function method for a flow in local equilibrium obeys the relations

$$U_p = \frac{U^*}{\kappa} \ln(EY_+), \tag{2}$$

$$k = \frac{U^{*2}}{\sqrt{C_\mu}}, \tag{3}$$

$$\varepsilon = C_\mu^{3/4} \frac{k^{3/2}}{\kappa Y}, \tag{4}$$

where $U_p$ is the absolute value of the velocity parallel to the wall at the first grid node, $U^*$ is the friction velocity calculated as $U^* = \sqrt{\tau_w/\rho}$, $\kappa$ is the von Karman constant equal to 0.41, $E$ is a roughness parameter dependent on the wall roughness taken equal to 8.6 for smooth walls, $Y_+$ is a dimensionless near wall quantity for length determined as $Y_+ = \frac{U^*Y}{\nu}$, $\nu$ is the turbulent viscosity, $Y$ is the distance of the first grid node to the wall and $C_\mu$ is a dimensionless constant equal to 0.09 in the standard $k - \varepsilon$ turbulence model. If the wall is considered to be rough (not smooth) then the roughness parameter $E$ is a function of

the Reynolds roughness number defined as $Re_r = \frac{U^*h_r}{\nu}$ where $h_r$ is the sand grain roughness height. The relation between the roughness parameter $E$ and Reynolds roughness number $Re_r$ follows the empirical laws proposed by Jayatilleke (1966):

$$E = 8.6 \text{ when } Re_r < 3.7, \tag{5}$$

$$E = \frac{1}{\sqrt{a(\frac{Re_r}{b})^2 + \frac{1-a}{8.6^2}}} \text{ when } 3.7 < Re_r < 100, \tag{6}$$

$$E = \frac{b}{Re_r} \text{ when } Re_r > 100, \tag{7}$$

where $b = 29.7$, $a = (1 + 2x^3 - 3x^2)$ and $x = 0.02248 \cdot (100 - Re_r)/Re_r^{0.564}$.

For the simulations no tower or hub effects are considered. The presence of the rotor is modelled using an actuator disc method based on the 1D momentum theory. The thrust force $F_i$ of each individual cell of the disc is calculated according to

$$F_i = C_T \left( U_{1,i} \right) \frac{1}{2} \rho \left( \frac{U_{1,i}}{1 - \alpha_i} \right)^2 A_i. \tag{8}$$

Where $U_{1,i}$ is the velocity of the flow at the individual cell numbered $i$ of the disc, $\alpha_i$ is the axial induction factor calculated for each individual cell of the disc, $A_i$ the surface area of the cell facing the undisturbed wind flow direction and $C_T \left( U_{1,i} \right)$ is a modified thrust coefficient curve dependent on the velocity at the disc. The modified thrust coefficient curve is created in a pre–processing step by replacing the undisturbed wind velocity values of the thrust coefficient curve with the wind velocity values at the disc $U_1$. To do this Eq. (9) is used, where $C_T$ is the thrust coefficient for the respective undisturbed wind velocity $U_\infty$.

$$U_1 = U_\infty \left( 1 - \frac{1}{2} \left( 1 - \sqrt{1 - C_T} \right) \right). \tag{9}$$

The total thrust force applied to the flow is calculated by summing the individual thrust forces according to $F_{tot} = \sum_i F_i$ over the disc area. This total thrust force may then be distributed in different ways over the disc. In this work, apart from using Eq. (8) as it is to prescribe the forces in each individual cell, referred to as the undistributed thrust, four different thrust distributions are tested: a uniform, a polynomial, a triangular and a trapezoidal distribution. Their equations are presented in Table 2, Fig. 2 presents a normalised plot of all four distributions along a diameter of the disc. The uniform distribution is chosen to match the thrust distribution of the actual rotor of this case. Full scale wind turbines however have a zero thrust value at the hub and at the tip of the blades, the polynomial distribution which is a fourth order polynomial is intended to respect this by having a zero thrust at the hub and at the tip of the disc. The triangular distribution is designed to have a zero thrust at the hub and to increase linearly the thrust force along the radius, up to the tip of the disc. Lastly, the trapezoidal distribution is set-up to resemble the thrust distribution produced using the actuator line method presented in Sarmast et al. (2012). While it is possible to determine a thrust distribution given the rotor geometry and airfoil data through a blade element momentum theory, it is somewhat impractical for industrial applications. Airfoil data of commercial wind turbines are generally not available to the typical industrial user. The purpose of testing different thrust distributions with this ACD method is that these will probably produce different wake properties e.g. with respect to the velocity deficit and turbulent kinetic energy of the wake. Two questions thus arise, i.e. which thrust distribution within this ACD method better captures the wake produced by a wind turbine and up to which distance does the thrust distribution have an effect on the wake? Here it should be noted that the primary goal is not to isolate the influence of the thrust distribution on the wake flow, as the total thrust over the disc will intrinsically vary depending on the thrust distribution used within the method. Here the goal is to investigate the effect the ACD method with different thrust distributions has on the wake flow.

For the first part of the simulations (case A) the numerical domain was defined according to the wind tunnel geometry as reported in Krogstad et al. (2011). Initially, empty domain simulations were conducted to assess the extent of unintended stream–wise gradients for the mean velocity and turbulence parameters. For this purpose horizontal profiles of $U$, $k$ and $\varepsilon$ are extracted at the inlet, turbine location and $x/R = 10$ downstream of the turbine position. As the roughness height of the wind

**Table 2.** Thrust distributions over the disc, where $r$ is the distance from the centre of the disc and $R$ is the radius of the disc. Note that $f$ on the left hand side of the equations has dimensions of force per unit area.

| Distribution | Equation | $b$ | Range of application |
|---|---|---|---|
| Uniform | $f_{uni,i} = bF_{tot} \frac{1}{\sum A_i}$ | $b = 1$ | $0 \le r \le R$ |
| Polynomial | $f_{pol,i} = bF_{tot} \left(\frac{r}{R}\right)^2 \left[1 - \left(\frac{r}{R}\right)^2\right] \frac{1}{\sum A_i}$ | $b = 6$ | $0 \le r \le R$ |
| Trapezoidal* | $f_{tra,i} = bF_{tot} \left(4\frac{r}{R} + 1\right) \frac{1}{\sum A_i}$ | $b = \frac{2}{7}$ | $0.2R \le r \le R$ |
| Triangular | $f_{tri,i} = bF_{tot} \left(\frac{r}{R}\right) \frac{1}{\sum A_i}$ | $b = \frac{3}{2}$ | $0 \le r \le R$ |

*For the Trapezoidal distribution there is no force applied in the region of $0 \le r < 0.2R$.

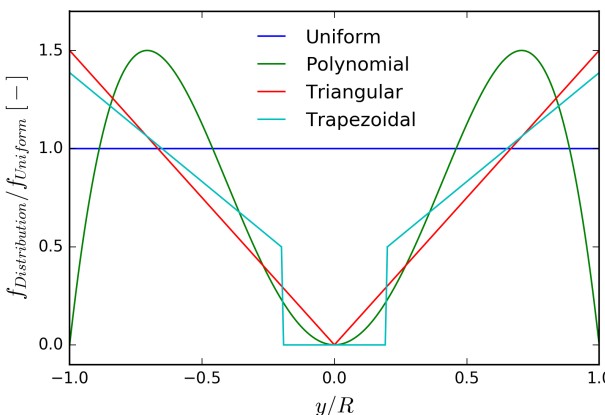

**Figure 2.** Normalised plot of all four distributions along a diameter of the disc.

tunnel walls is not known a priori, a comparison between the experimental boundary layer profile and the simulated boundary layer profile for different roughness height values is conducted in a trial and error fashion until the appropriate value for the roughness height is found. When considering the wind turbine in the simulation the computed results are compared against the HWA measurements for the normalised axial velocity $U/U_{ref,A}$ and normalised turbulent kinetic energy $k/U_{ref,A}^2$ at the three
5  downstream positions mentioned in Table 1 along the horizontal line through the centre of the wake in the crosswise direction.

For case B the domain geometry and the positioning of the wind turbines are in accordance with the invitation sent out by Pierella et al. (2012). The equilibrium wall function method for smooth walls, that is $E = 8.6$, is used to introduce the effects of the wind tunnel walls into the numerical simulation, this applies as well for case C. The computed results when the ACDs are considered are compared against HWA measurements for the normalised axial velocity $U/U_{ref,B}$ and normalised variance of
10  the axial velocity component $\overline{u'^2}/U_{ref,B}^2$ at the three downstream positions shown in Table 1 along the horizontal line through the centre of the wake in the crosswise direction. Further the thrust coefficients $C_T = \frac{2F_{tot}}{\rho U_{ref,B}^2 A}$ of the two wind turbines are compared with the experimental results, where $A$ is the rotor cross section of each individual wind turbine. Here it should be

noted that even though the thrust coefficient curve is an input to the simulation, the thrust coefficient value applied against the flow depends upon the velocity at the disc which changes as the simulation progresses.

Lastly, for case C the domain geometry and the positioning of the wind turbines are in accordance to the invitation sent out by Sætran and Bartl (2015). Prior to the simulations with the ACD, empty domain simulations are performed for the sub–cases (C1, C2). This is to match the inlet wind profile and turbulence intensity with the experimental measurements at four downstream positions from the inlet, that is $x/R = 4.00$, $9.54$, $14.36$ and $22.00$. When considering the wind turbines in the numerical simulation via the ACD method, the computed results are compared with the HWA measurements for the normalised axial velocity $U/U_{ref,C}$ and normalised turbulent kinetic energy $k' = k/U_{ref,C}^2$, where $U_{ref,C} = 11.5 \text{ m s}^{-1}$ . The thrust coefficients of the two turbines is calculated as $C_T = \frac{2F_{tot}}{\rho U_{ref,C}^2 A}$.

## 2.3  Grid convergence analysis

A grid independence study is carried out according to the recommended procedure of Roy (2003) for mixed order schemes. For these simulations a uniform grid is used based on the cells per rotor diameter. Table 3 presents information regarding the different grid levels used in the grid independence study. Even though the grid independence study is performed solely for case A it is considered to apply for the other cases as well.

**Table 3.** Grid levels and size.

| Grid level | Cells per rotor diameter | Cells in the domain |
|---|---|---|
| 1 | 40 | $48 \times 10^5$ |
| 2 | 20 | $6 \times 10^5$ |
| 3 | 10 | $3 \times 10^5$ |

According to Roy (2003) the series that represents the discrete solution for each grid is given by

$$f_k = f_{exact} + g_1 h_k + g_2 h_k^2 + \mathcal{O}(h_k^3). \tag{10}$$

Where $f_k$ is the discrete value solution of grid k, $g_i$ is the i–th order error term coefficient and $h_k$ is a measure of the grid spacing. The three unknowns ($f_{exact}$, $g_1$ and $g_2$) may be found by expanding Eq. (10) for three consecutive grids and by solving the resulting three set equation.

$$f_1 = f_{exact} + g_1 h_1 + g_2 h_1^2 + \mathcal{O}(h_1^3), \tag{11}$$

$$f_2 = f_{exact} + g_1 h_2 + g_2 h_2^2 + \mathcal{O}(h_2^3), \tag{12}$$

$$f_3 = f_{exact} + g_1 h_3 + g_2 h_3^2 + \mathcal{O}(h_3^3). \tag{13}$$

**Table 4.** Overview of the cases for which results will be presented.

| Case | Grid and wall function study | Thrust distribution used for different turbulence models | Turbulence model used for different thrust distributions | Thrust coefficient comparison | Empty domain study |
|------|------|------|------|------|------|
| A | yes | Uniform | $k - \varepsilon$ | – | – |
| B | – | Undistributed | $k - \varepsilon$ | yes | – |
| $C_1$ | – | Undistributed | $k - \varepsilon$ | yes | yes |
| $C_2$ | – | Undistributed | $k - \varepsilon$ | yes | yes |

The spatial discretization error is calculated according to the following formula

$$|\text{spatial error}(\%)| = \left| \frac{f_k - \widetilde{f}_{exact}}{f_{exact}} \right| \times 100. \tag{14}$$

Where $f_k$ is the discrete value solution of grid k and $\widetilde{f}_{exact}$ is an approximation to the exact solution $f_{exact}$ which is found by disregarding the higher order terms of Eq. (11) to Eq. (13). The normalized magnitudes of the first and second order error terms and the magnitude of their sum (mixed order) is given by

$$\left| \frac{\widetilde{g}_1 h}{\widetilde{f}_{exact}} \right| \times 100, \qquad \left| \frac{\widetilde{g}_2 h^2}{\widetilde{f}_{exact}} \right| \times 100, \qquad \left| \frac{\widetilde{g}_1 h + \widetilde{g}_2 h^2}{\widetilde{f}_{exact}} \right| \times 100. \tag{15}$$

Where $\widetilde{g}_1$ and $\widetilde{g}_2$ are approximations to $g_1$ and $g_2$ which are found as mentioned previously by solving and disregarding the higher order terms of Eq. (11) to Eq. (13).

## 3 Results and Discussion

A summary of the cases for which results will be presented is shown in Table 4. For cases A and B, empty domain results for the axial velocity extracted at cross-sectional horizontal profiles at the inlet, turbine location and at a position $x/R = 10$ downstream of the turbine location, show approximately a 2.7 % increase of the axial velocity. The turbulence parameters $k$ and $\varepsilon$ on the other hand, decrease steadily from the inlet to $x/R = 10$, which is due to the lack of a turbulence generating mechanism along the domain e.g. shear. The decrease in $k$ and $\varepsilon$ along the empty domain is five orders of magnitude lower than their average value when an ACD model is present in the computations.

Figure 3 presents the spatial discretisation error results obtained for the normalised axial velocity profiles at three distances downstream of the wind turbine position for case A. The error is estimated to be less than 2.4 % for finest grid (Grid 1). Therefore for the purpose of this investigation a uniform grid resolution of 40 cells per rotor diameter is found suitable for all cases. Also shown in Fig. 3 are the normalized magnitudes of the first and second order error terms and of their sum, which are given by Eq. (15).

Figure 4 illustrates the computed stream–wise velocity results for two different wall function values, against velocity measurements conducted in the wind tunnel with pitot tubes. A quite good match between experimental measurements and the

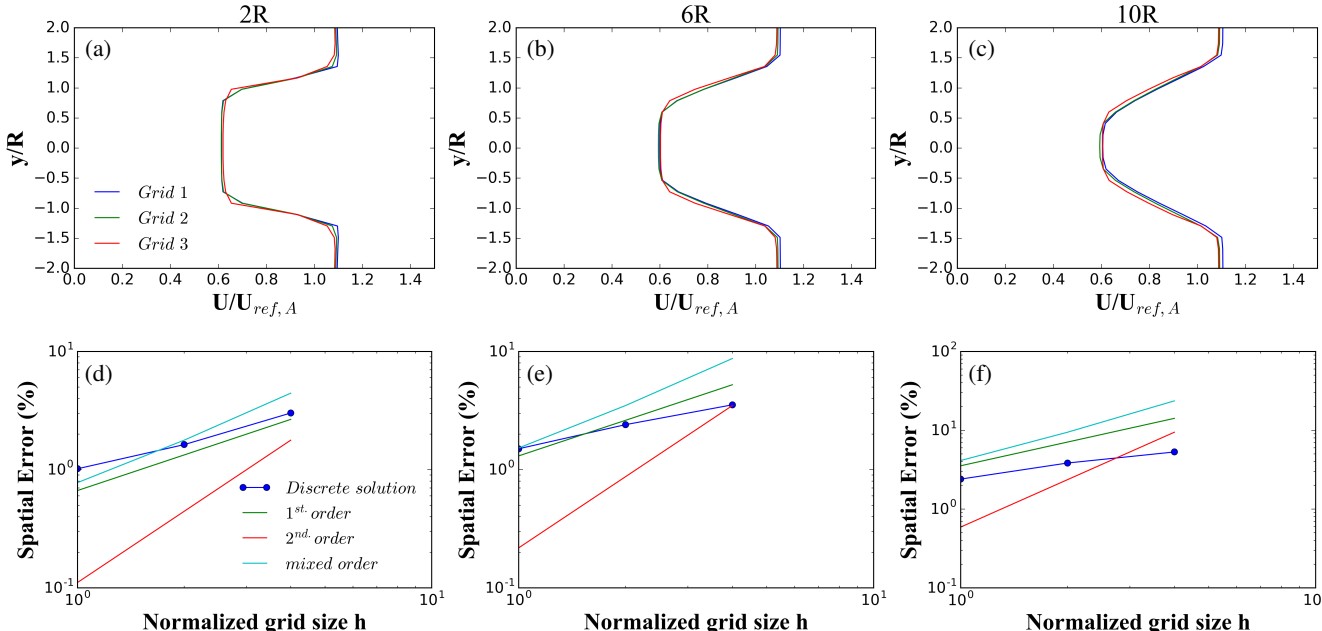

**Figure 3.** Normalised axial velocity and spatial discretization error computed behind a single model wind turbine (case A) for three grids at (a),(d) x/R=2, (b),(e) x/R= 6 and (c),(f) x/R=10 for the $k-\varepsilon$ turbulence model and the undistributed distribution.

computed results exists when considering a smooth wall i.e. $E = 8.6$. Therefore, the equilibrium wall function for smooth walls is used in the simulations, as it is found to have the best agreement with the measurements. By setting a sand grain roughness height other than that for a smooth wall causes the discrepancy between the simulated boundary layer development and the measurements to increase for the axial velocity profile. Figure 5 illustrates the normalised axial velocity contours for case A and B. The polynomial thrust distribution is used along with the $k-\varepsilon$ turbulence closure model. It is clearly seen that the method reproduces what is expected, that is by positioning a second turbine in the wake of the first the axial velocity of the flow is further reduced. This reduction is due to the further energy extraction of the second wind turbine from the mean flow. The dashed lines in Fig. 5 indicate the positions at which flow values are extracted and compared with the HWA measurements.

For case A, the computed results are validated against HWA measurements for the normalised axial velocity and normalised turbulent kinetic energy, see Fig. 6. These results are computed using different turbulence models and the uniform thrust distribution. To investigate the influence of the thrust distribution on the wake development, simulations using the $k-\varepsilon$ turbulence model with different thrust distributions were conducted, results are shown in Fig. 7.

In Fig. 6(a) it is observed that the $k-\varepsilon$ and the $KL\ k-\varepsilon$ turbulence models produce results similar to the measurements, with the $KL\ k-\varepsilon$ model being less diffusive than the $k-\varepsilon$ model in the crosswise direction. Apart from the undistributed thrust, all thrust distributions used in this study assume axisymmetry, therefore the simulated profiles are symmetrical to the rotor centre. However, this is not the case with the measurements, which exhibit asymmetric profiles as seen in Fig. 6(a). According to Krogstad and Eriksen (2013) this asymmetry may be produced by the slowly rotating tower wake as seen e.g. at

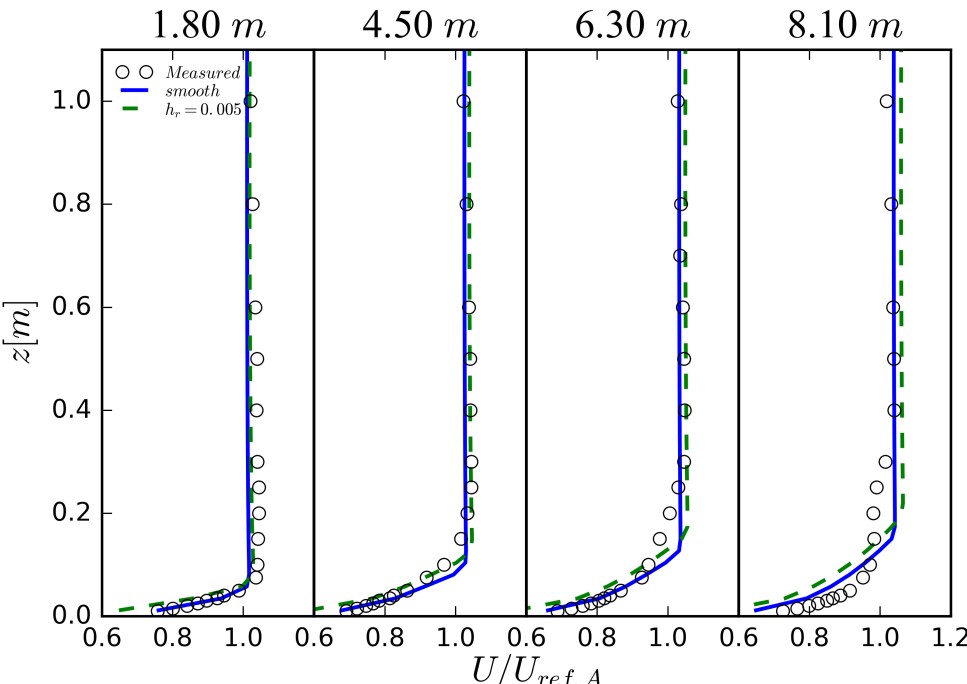

**Figure 4.** Normalised axial velocity for different wall functions plotted against the experimental data for four downstream distances from the inlet.

the downstream position of $x/R = 10$. As the ACD is non-–rotating in the simulations it is expected that the predictions will not capture this asymmetry in the wake. Further, as the effects of the nacelle and tower are not considered it is also anticipated to find small deviations of the predictions with the measurements in the near wake. On average the blockage effect is captured by the simulations as seen in Fig. 6(a). This effect is apparent outside the wake region ($|y/R| > 1.5$) where the simulated and

5   measured normalised axial velocity values are higher than one. Considering the normalised turbulent kinetic profiles in Fig. 6(b), the shape of the profiles is not successfully predicted by any of the turbulence models. The $k-\omega$ turbulence model tends to over-predict the turbulent kinetic energy production in this low background turbulence environment. As a result the simulated wake recovery in comparison to the velocity measurements is too high. The discrepancy between the measured profile shape of the turbulent kinetic energy from that predicted at the downstream position of $x/R = 2$ for the wake region of $|y/R| < 0.5$

10  is mainly due to the presence of the nacelle and abrupt change of the blade shape from the airfoil profile to a cylinder near the nacelle.

When keeping the turbulence model constant and changing the thrust distribution it is observed in Fig. 7 that the effect of the thrust distribution is pronounced in the near wake region and diminishes further downstream. Porté-Agel et al. (2011) also observed that the effect of representing the forces of a wind turbine differently, such as by a rotating or non-rotating

15  ACD or an ACL was more pronounced in the near wake region, rather than in the far wake region. Regarding the turbulence

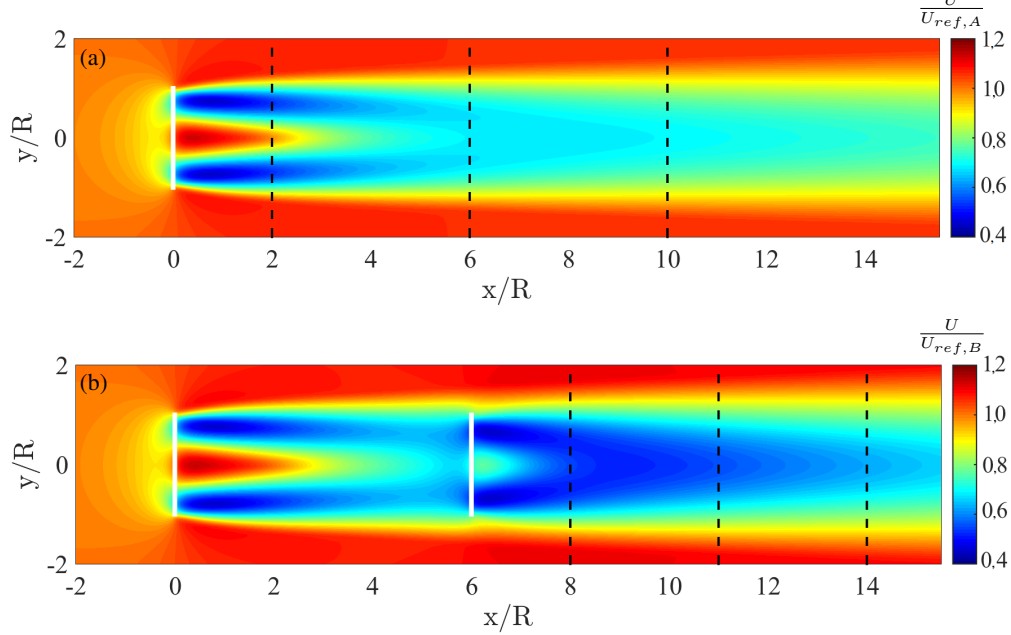

**Figure 5.** Normalised axial velocity contours for the $k - \varepsilon$ turbulence model using the polynomial thrust distribution. (a) One model wind turbine, case A and (b) two in-line model wind turbine, case B.

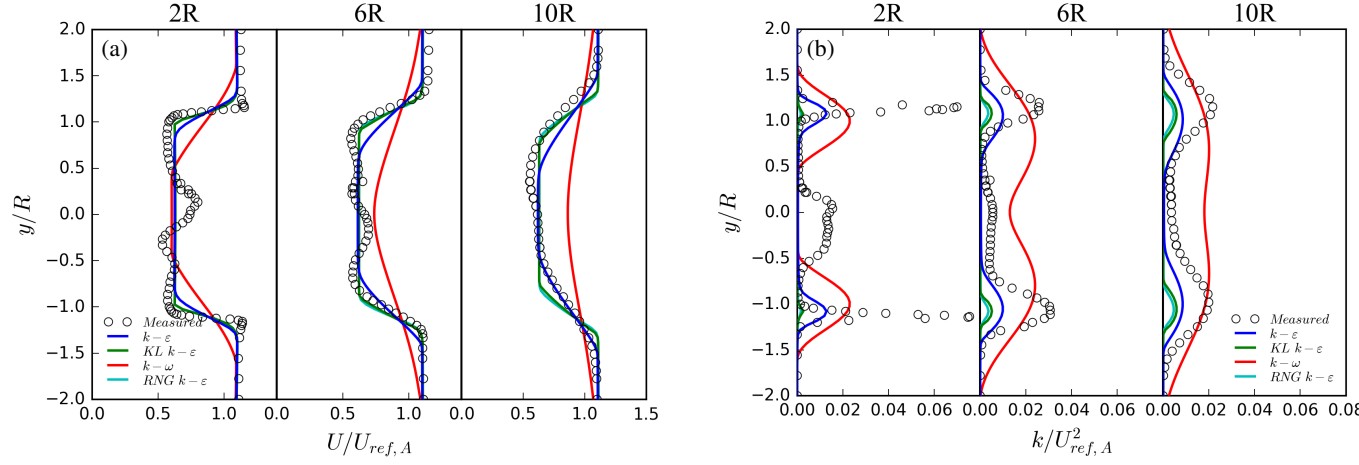

**Figure 6.** (a) Normalised axial velocity and (b) normalised turbulent kinetic energy computed behind a single model wind turbine (case A) at $x/R = 2$, 6 and 10 for different turbulence models using the uniform distribution.

kinetic energy, all thrust distributions seem to capture the position of the tip vortex apart from the polynomial. The increased turbulence production due to the breakdown of the tip vortex at the $x/R = 2$ position is not captured by any combination of thrust distribution and turbulence model. This is also observed by other researchers such as Réthoré et al. (2014), which

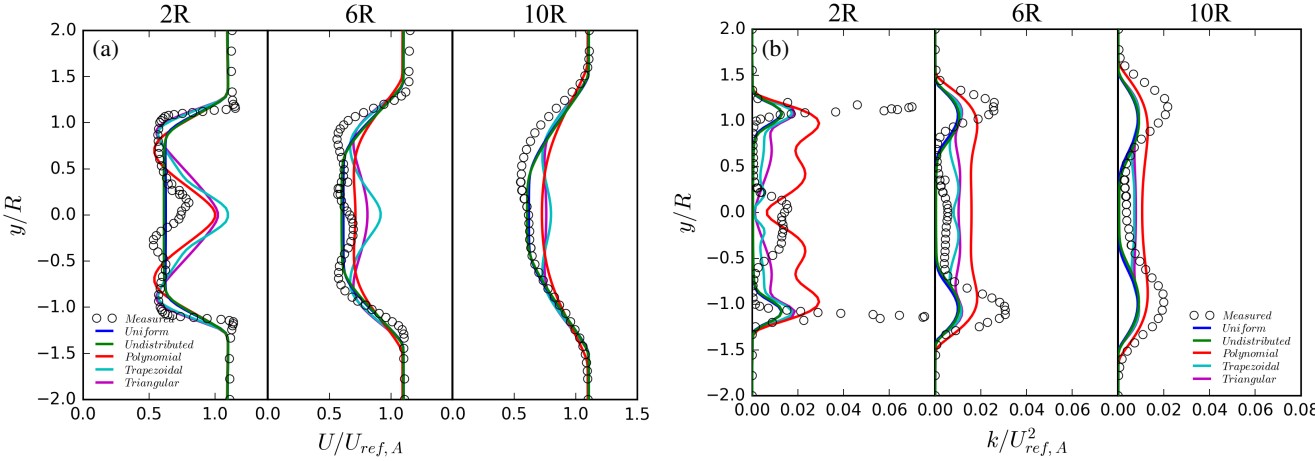

**Figure 7.** (a) Normalised axial velocity and (b) normalised turbulent kinetic energy computed behind a single model wind turbine (case A) at $x/R = 2$, 6 and 10 for different thrust distributions and the $k - \varepsilon$ turbulence model.

concluded that the ACD method lacks the ability to simulate the turbulent structures present in the near wake region. Sumner et al. (2013) results also show that in a low background turbulence intensity environment there seemed to be a perceptible dependency of the wake development on the turbulence closure used, in terms of velocity deficit and turbulent kinetic energy.

For case B when using the undistributed thrust, the normalised axial velocity and stream–wise variance of the velocity at three positions downstream of the second turbine are shown in Fig. 8 for different turbulence models. In Fig. 9, the effect of the thrust distribution on the wake downstream of the second wind turbine is investigated by varying the thrust distribution while keeping the same turbulence model. Results of the thrust coefficient values for the upstream and downstream wind turbines are summarised in Table 5 for when the $k - \varepsilon$ turbulence model is used.

For case B in which the wakes of both wind turbines interact, similar to the results of case A, the $k - \varepsilon$ and the $KL$ $k - \varepsilon$ turbulence models produce axial velocity results in agreement with the measurements. The $KL$ $k - \varepsilon$ seems however to underestimate the normalised stream–wise variance of the velocity. The $k - \omega$ turbulence model here again over–predicts the wake recovery and over estimates the normalised stream–wise variance of the velocity. On the contrary the $RNG$ $k - \varepsilon$ under–predicts the wake recovery and under estimates the normalised stream–wise variance of the velocity. The blockage effect is on average captured for all turbulence models, apart from when the $k - \omega$ turbulence model is used, and thrust distributions as seen in Fig. 8 and Fig. 9. The wake expansion is accurately predicted when using the $k - \varepsilon$ turbulence model. Further, when keeping the turbulence model constant and changing the thrust distribution (Fig. 9) it is observed that the effect of the thrust distribution is less pronounced in the two in–line wind turbine case than for the single wind turbine case due to the higher turbulent diffusion. The wake predicted when using the undistributed or uniform thrust distribution seems to be in closer agreement to the measurements. This is possibly due to the fact that these distributions produce a fairly constant pressure drop over the disc as do the blades when operating at their design condition. The thrust coefficient values summarised in Table 5 for

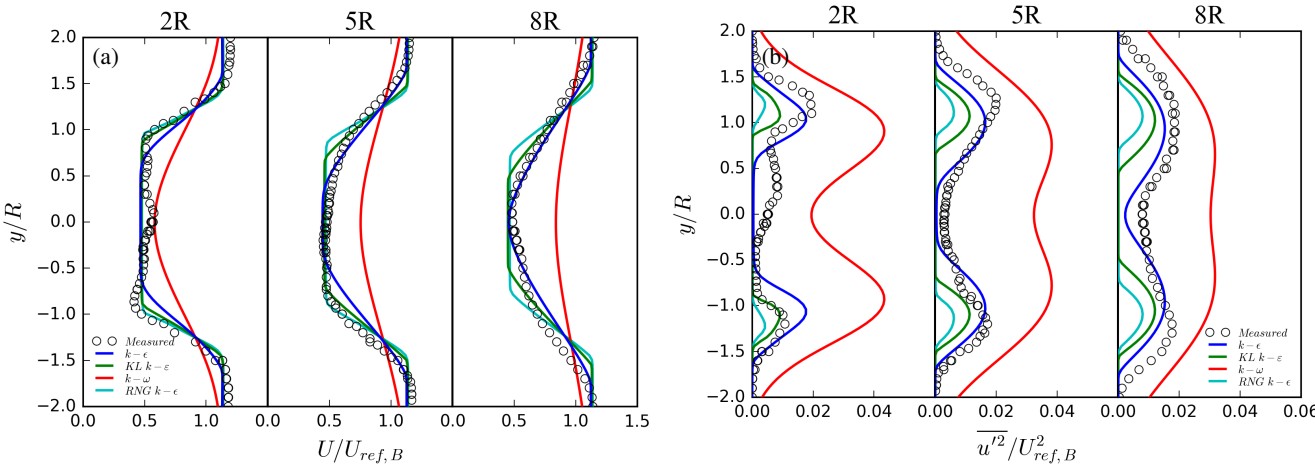

**Figure 8.** (a) Normalised axial velocity and (b) normalised stream–wise variance of the velocity computed for the two in-line model wind turbines (case B) at $x/R = 2$, 5 and 8 downstream of the second wind turbine for different turbulence models using the undistributed thrust.

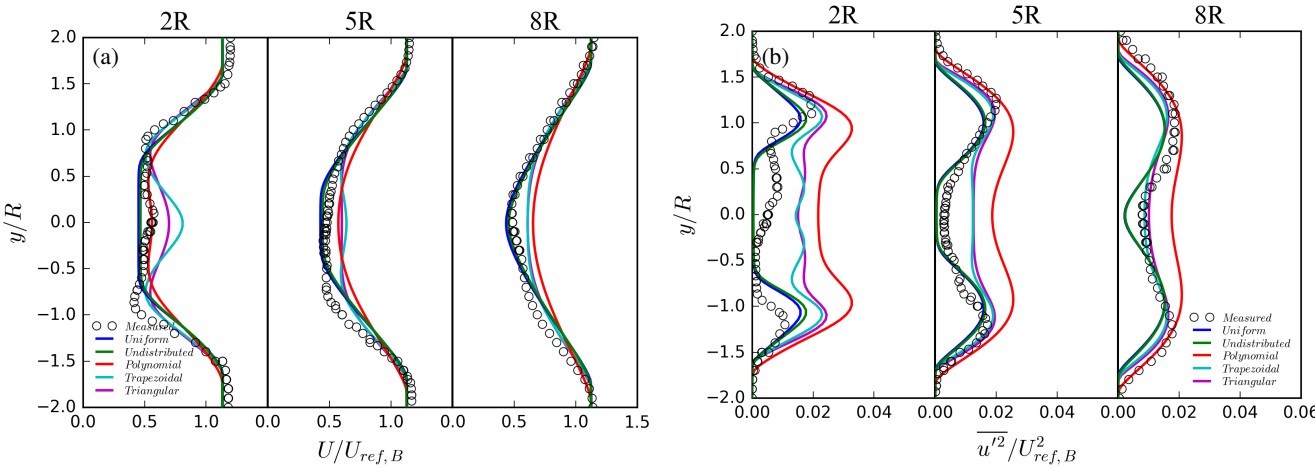

**Figure 9.** (a) Normalised axial velocity and (b) normalised stream–wise variance of the velocity computed for the two in-line model wind turbines (case B) at $x/R = 2$, 5 and 8 downstream of the second wind turbine for different thrust distributions and the $k - \varepsilon$ turbulence model.

the different thrust distributions and the $k - \varepsilon$ turbulence model agree quite well with the measured data. There is, on average, a 5 % difference between the measured thrust and the results for the upstream wind turbine and less than a 10 % difference for all cases concerning the downstream wind turbine with the exception of the results when using the undistributed or uniform thrust. These differences increase when considering different turbulence models, this is due to the different associated wake 5 development corresponding to the different turbulence models. When considering the $k - \omega$ turbulence model the simulations greatly overestimate the values of the thrust coefficient for the second wind turbine and vice versa for the $RNG \ k - \varepsilon$.

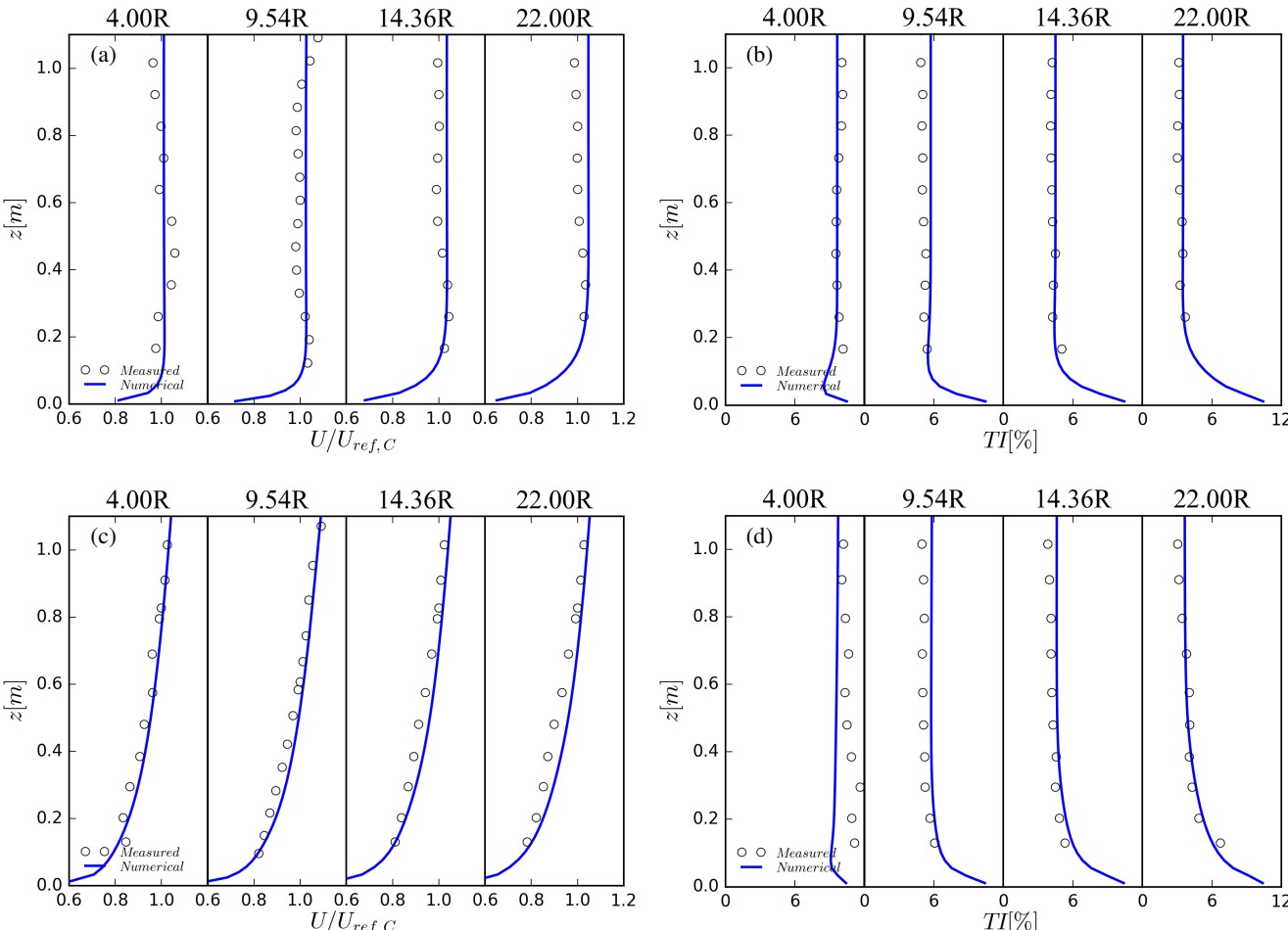

**Figure 10.** Empty domain stream–wise velocity and turbulence intensity results for Case C1 (a),(b) and Case C2 (c),(d) for four vertical profiles downstream of the inlet when using the $k - \varepsilon$ turbulence model.

Concerning sub–cases C1 and C2, empty domain vertical profile results for the stream–wise velocity and the turbulence intensity at four axial positions downstream of the inlet are shown in Fig. 10. Illustrated in Fig. 11 are the measured and simulated axial velocity and turbulent kinetic energy at three distances downstream of turbine $T_1$ when using the undistributed thrust and different turbulence models for case C1. Fig. 12 presents the same quantities when using the $k - \varepsilon$ turbulence model but varying the thrust distribution. Lastly, for case C2 results for the stream–wise velocity and the turbulence intensity profiles are shown in Fig. 13 for different turbulence models when using the undistributed thrust. The effect of the thrust distribution on the wake is investigated by varying the thrust distribution while keeping the same turbulence model, see Fig. 14. Results of the thrust coefficient values for the upstream and downstream wind turbine are summarised in Table 5 for cases C1 and C2, the $k - \varepsilon$ turbulence model is used here.

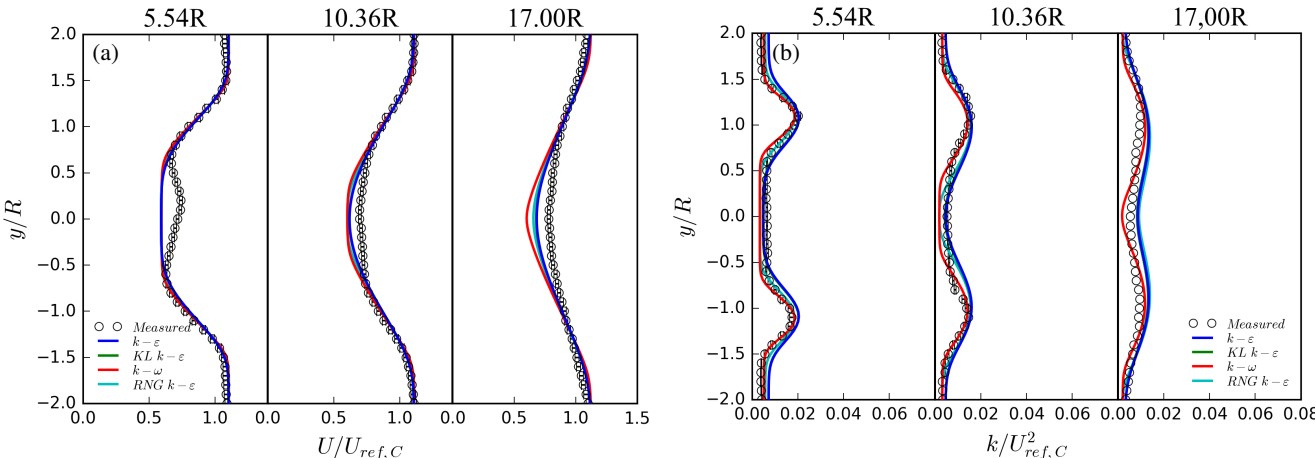

**Figure 11.** (a) Normalised axial velocity and (b) normalised turbulent kinetic energy computed for the two in-line model wind turbines (case C1) at $x/R = 5.54$, 10.36 and 17.00 downstream of the first wind turbine for a separation distance of $18.00R$. Different turbulence models are used along with the undistributed thrust.

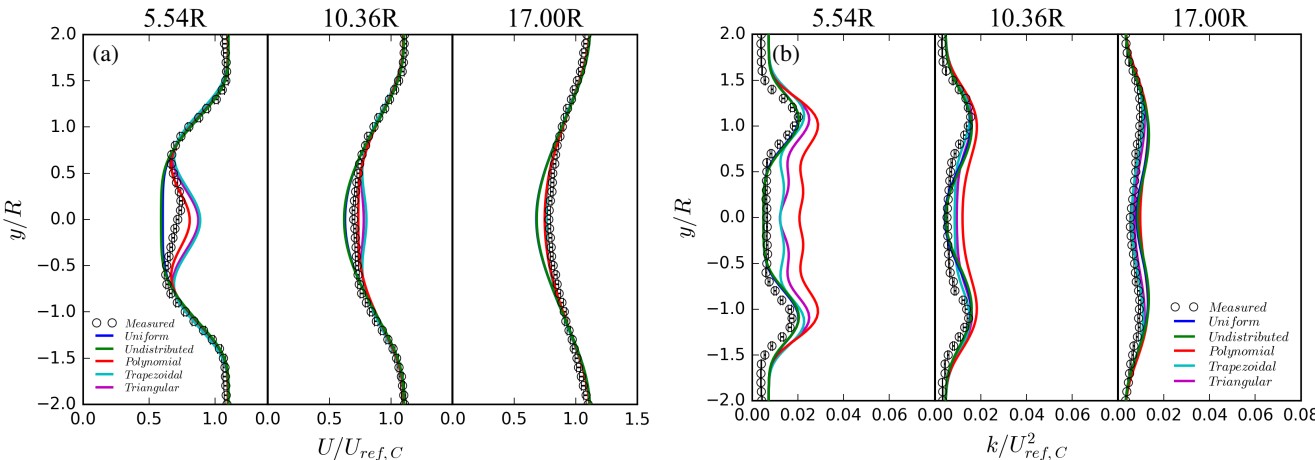

**Figure 12.** (a) Normalised axial velocity and (b) normalised turbulent kinetic energy computed for the two in-line model wind turbines (case C1) at $x/R = 5.54$, 10.36 and 17.00 downstream of the first wind turbine. Different thrust distributions are used and the $k - \varepsilon$ turbulence model.

The empty domain simulations presented in Fig. 10 give a reasonably good agreement between simulations and measurements for both the uniform and the sheared inflow condition. Concerning the sub-case C1, the simulated stream–wise velocity is in quite good agreement with the measurements for all turbulence models and thrust distributions, see Fig. 11 and Fig. 12. In this high background turbulence environment case the $k - \omega$ turbulence model shows remarkably better agreement with the measurements when compared to the previous cases. The shape and level of the normalised turbulent kinetic energy is

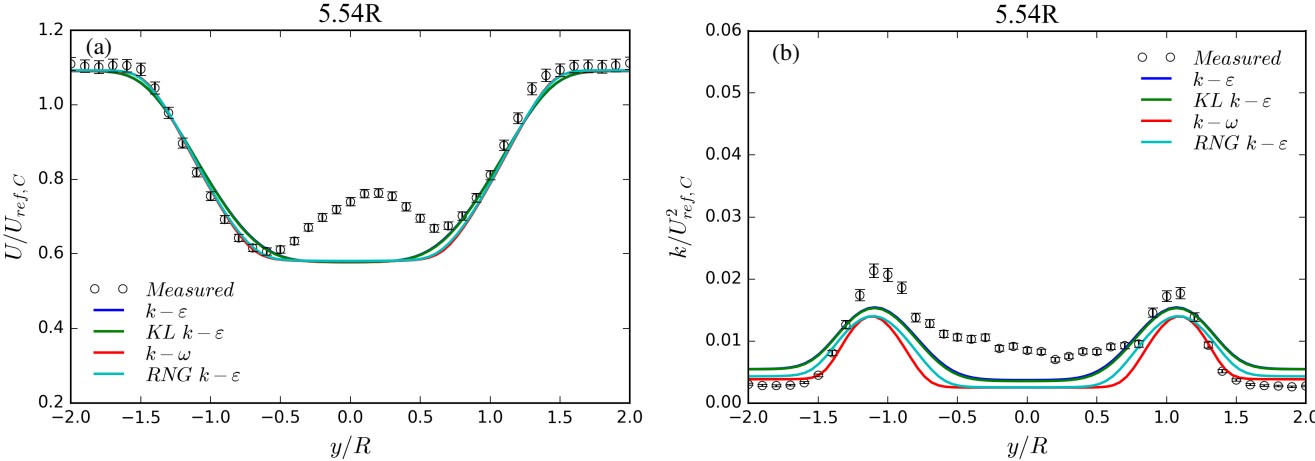

**Figure 13.** (a) Normalised axial velocity and (b) normalised turbulent kinetic energy computed for the two in-line model wind turbines (case C2) at x/R=5.54 downstream of the first wind turbine. Different turbulence models are used along with the undistributed thrust.

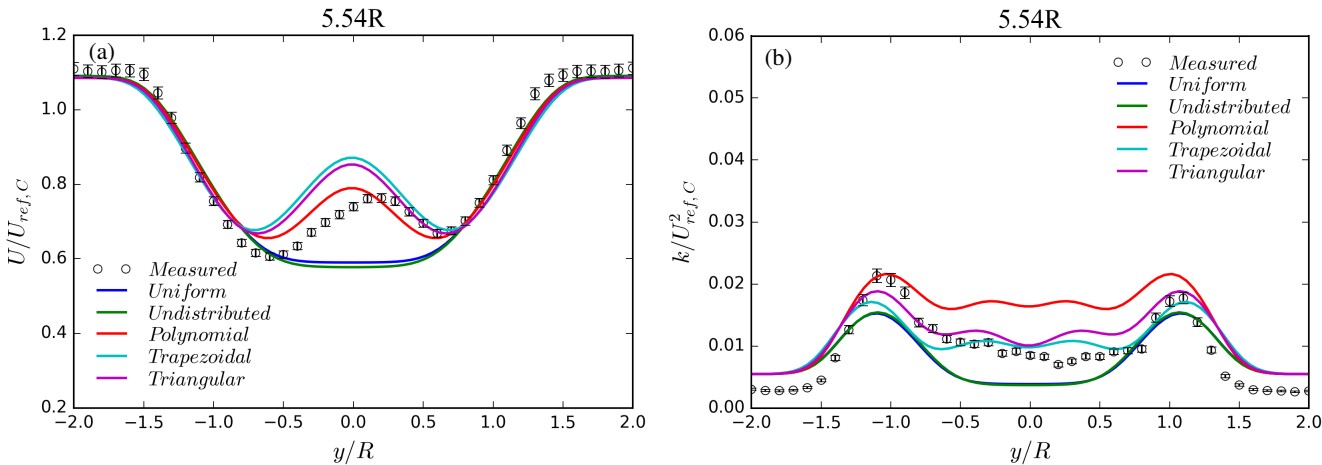

**Figure 14.** (a) Normalised axial velocity and (b) normalised turbulent kinetic energy computed for the two in-line model wind turbines (case C2) at x/R=5.54 downstream of the first wind turbine. Different thrust distributions are used and the $k - \varepsilon$ turbulence model.

now captured by all turbulence models though small differences are observed between the simulated wakes when different turbulence models are used. This finding is in agreement with results from the studies performed by Laan et al. (2015b) and Sumner et al. (2013), where small differences between the simulated wakes are also found when different RANS turbulence models are used in a high background turbulence intensity environment. It should be recalled that empty domain simulations 5 were performed to match the background turbulence intensity with the experimental measurements when using different turbulence models for all cases prior to the simulations with the ACD. It appears that for the high turbulence intensity cases this

**Table 5.** Thrust coefficients for case B and sub–cases C1 and C2 when using the $k-\varepsilon$ turbulence model for the first and second wind turbine.

|  | Case B | | Sub–case C1 | | Sub–case C2 | |
|---|---|---|---|---|---|---|
|  | $C_{T,T_1}$ | $C_{T,T_2}$ | $C_{T,T_1}$ | $C_{T,T_2}$ | $C_{T,T_1}$ | $C_{T,T_2}$ |
| Experimental | 0.883 | 0.363 | 0.833 | 0.569 | 0.785 | 0.486 |
| Polynomial | 0.797 | 0.379 | 0.825 | 0.608 | 0.809 | 0.454 |
| Trapezoidal | 0.825 | 0.367 | 0.856 | 0.621 | 0.841 | 0.478 |
| Triangular | 0.822 | 0.390 | 0.837 | 0.600 | 0.824 | 0.457 |
| Uniform | 0.830 | 0.269 | 0.854 | 0.571 | 0.840 | 0.396 |
| Undistributed | 0.829 | 0.268 | 0.853 | 0.580 | 0.840 | 0.400 |

procedure has a significant effect on the computational results compared to the low turbulent intensity cases. As the purpose of using different turbulence models in this study is to investigate the effect of the turbulence model with its defined constants on the wake development. It is crucial to set throughout the domain the background turbulence intensity in accordance to the experimental set–up when using different turbulence models. This is achieved here by varying the inlet turbulent parameters ($k$, $\varepsilon$ or $\omega$), in this way the background turbulence intensity is similar when using different turbulence models and the effect of the turbulence model on the wake development may be clearly accounted for. When the higher background turbulence intensity cases are considered, it is observed that the effect of the thrust distribution is less apparent further downstream compared to the lower turbulence intensity of cases A and B. Similarly, for sub–case C2 (see Figs. 13 and 14) the axial velocity in the wake is predicted quite well for all turbulence models and thrust distributions. Because the measurement position is in the near wake region the effect of the thrust distribution is apparent on the turbulent kinetic energy and axial velocity profile. From Table 5 it is found that thrust coefficient values are estimated on average to have less than a 10% difference from the measured values for both sub–cases when using the $k-\varepsilon$ turbulence model.

Lastly, in terms of computational time or CPU hours, herein defined as the number of CPUs $\times$ wall clock time needed to perform the simulation, results are shown in Table 6. These results present the CPU hours needed to perform the simulations using this method and a LES with the actuator line method described in Sørensen et al. (2015) for case A. It is found that the RANS/ACD method is significantly faster in simulating this one wind turbine case compared to the LES/ACL method. Although the LES/ACL method provides high fidelity results comparable to the measurements, the computational requirements of this method, up to this day, are still too demanding to make it usable for wake modelling in industrial applications.

## 4 Conclusion

The main conclusions of this study are summarised as follows: (i) the present results, considering the simplicity and low computational needs of the method, generally show satisfactory agreement between the simulations and the measurements used

**Table 6.** Averaged computational effort in CPU hours to perform the simulations of case A.

| Description | Cells per rotor diameter | Cells in the domain | CPU hours |
|---|---|---|---|
| RANS ACD | 40 | $4.8 \times 10^6$ | 20 |
| LES ACL | 86 | $24.5 \times 10^6$ | 1280 |

for both the one wind turbine set-up (case A) and the two in–line wind turbine set–up (cases B and C). (ii) The effect of using different thrust distributions on the profiles is generally present in the near wake and fairly absent in the far wake. Moreover, the impact on the near wake is more pronounced for the single wind turbine set–up than in the two wind turbine set–up. (iii) The uniform and undistributed thrust distributions generally outperformed the other distributions in terms of the estimated wake.

Please note however that the uniformly distributed thrust might not be the best suited when considering near wake effects, if a full size wind turbine is modelled that typically has a non-uniform thrust distribution. (iv) Changing the turbulence model has a noticeable impact on the wake development in the low background turbulence intensity cases. When using the $k - \varepsilon$ and $KL$ $k - \varepsilon$ turbulence models the velocity results are in agreement with the measurements, but this is generally not the case with the $k - \omega$ turbulence closure models. (v) When considering the high background turbulence intensity cases, small differences in the

wake development are found by changing the turbulence model. The results however are quite sensitive to the inlet conditions of the turbulent parameters used for the simulations. Therefore depending on the turbulence model, the turbulent parameters at the inlet should be carefully considered as to represent the background turbulent experimental conditions. (vi) The wake in terms of the velocity and turbulence profiles was captured more accurately in the high background turbulent cases than in the low ones.

This method has shown to give reliable results for a number of different wind flow conditions and separation distances with respect to the single and the two in-line wind turbine cases. However it has not been validated yet for wind turbines operating in a situation where only a part of the rotor is in the wake of the upstream wind turbine (partial wake situation). Moreover, it has also not been validated against operational data measured within existing wind farms operating in full scale atmospheric conditions. Therefore, future research will focus on validating the method against data retrieved from operating wind farms.

Cases in which wind turbines are operating partially in the wake of the upstream turbine will be of special interest as well.

## 5   Data availability

The data presented in this article can be obtained by contacting the corresponding author.

*Competing interests.*   The authors declare that they have no conflict of interest.

*Acknowledgements.* This work is financially supported by the Research Council of Norway (Project no. 231831). The authors would also like to thank the Norwegian Technical National University and especially Jan Bartl for providing the experimental data. Andrew Barney is also gratefully acknowledge for assisting in the proof-reading of the manuscript.

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
