# Peer review of "Validation of the actuator disc approach using small scale model wind turbines"

_Wind Energy Science, 2017_

## Referee Comment (RC2)

Review of
**"Validation of the actuator disc approach using small scale model wind turbines"**
by N. Simisiroglou et al.
(wes-2017-18)

June 24, 2017

**1. General comments**

The paper compares wake flow and thrust predictions by an actuator disc approach with experimental data from NTNU's Blind test experiments. Specifically, the performance of four different turbulence closure models and five radial thrust distributions on the actuator disc are investigated. The model turbines' performance is analyzed for a single and double turbine setup and different turbulent inflow conditions.

Given the advantages of lower computational effort compared to LES/DES models or fully-resolved RANS model, this approach is considered to be relevant for a fairly accurate modeling of wind farm flows.

The paper follows an elaborate line of reasoning and brings up a number of clear conclusions. However, the manuscript is missing some crucial elements. A comprehensive literature review on the field of wake modeling is not included, neither are the results discussed with respect to the state-of-the-art in wake modeling Due to the lack of a wider context the current manuscript does not yet clearly demonstrate the advantages of the chosen modeling approach over other methods.

**2. Specific comments**

1. Firstly, I do not completely agree with the chosen structure of the manuscript. The content of chapters "3 Results" and "4 Discussion" is complicated to follow in the current structure.

   (a) I would suggest creating a new chapter 3 called "Precursor simulations" in which the empty tunnel simulations as well as the grid independency study are shortly described. The empty tunnel simulations (Fig. 3 and Fig. 9) as well as the grid independency study (Fig. 2) are rather boundary conditions than actual results in my opinion. Probably, it is not even necessary to show all modeled and measured inflow profiles rather than

shortly mentioning that the match is very good (and stating a deviation in %).

(b) In my opinion it would be more straightforward to discuss the results in the actual "results" chapter rather than separating results and their discussion.

(c) A "discussion" chapter, however, still would be essential to include. Therein, the main findings should be discussed with respect to previous findings in the literature. So far, there is only one reference (Laan et al., 2015) included in the discussion, but there is a huge variety of publications dealing with numerical wake simulations by now. It would add great value to the manuscript to discuss the observed effects with respect to other simulations (ACD, but also other RANS approaches (ACL or fully-resolved)).

(d) It could be useful to create a new chapter "Conclusions", starting from l.34 on p.16.

2. Secondly, several aspects of the thrust modeling require some deeper explanation. As the variation in radial thrust distribution is one of the two major parameters varied in this study an in-depth explanation of its modeling is deemed to be crucial

(a) A more elaborate description of the choice of thrust distributions and the associated parameters is needed. A plot showing $C_T$ (or $a$) vs. $r/R$ comparing the distributions given by the equations in Table 2 would help to illustrate the approach. How are the parameters b (Table 2) chosen?

(b) The distribution of the axial induction (or thrust) is not necessarily uniform along the rotor radius, depending on the rotor design and operational state. However, it should be possible to calculate radial distribution of the axial induction factor $a$ and thus the thrust for a given rotor design and operating point. A simple Blade Element Momentum code or turbine modeling tool (FAST, QBlade, ...) should do the job, if the rotor geometry and airfoil polars are available. To my understanding it thus should be possible to define a thrust distribution and eliminate it as a variable.

(c) Furthermore, It is not clear to me, how the downstream turbine's thrust coefficient $C_{T,T2}$ is calculated in cases B and C. Is it calculated from the fluid-ACD interaction or is the experimental $C_{T,T2}$ value used as an input? Please elaborate on the very short explanation given in l.5, p.7. See also comment 3 (c).

(d) Can you elaborate on what is meant by "undistributed" thrust? I did not find an explanation on that.

3. Finally, the scientific contribution of this work to the field of numerical wind turbine wake modeling should be stated in a clearer way.

(a) Elaborate in the introduction why you chose the presented modeling approach. What is the advantage of RANS-ACD modelling compared to

other numerical modeling techniques (LES/DES, ACL, ...)? Can you present some numbers justifying this approach with respect to computational effort (time)? Would this modeling approach thus have significant advantages in the modeling of a full wind farm?

(b) As stated in comment 1 (c) already, a discussion of the presented results with respect to state-of-the-art numerical wake modeling is deemed to be crucial. A discussion of both the approach and results by referring to other simulations would set this work into a broader context.

(c) The presented simulations of mean velocity and turbulent kinetic energy show very promising results, especially those simulated in a highly turbulent environment. However, I do not understand how the upstream turbine's $C_{T,T1}$ and especially the downstream turbine's thrust coefficient $C_{T,T2}$ are dependent on experimental information. For the modeling of a bigger wind farm, it would be important to be able to calculate the coefficients based on the information given in turbine data sheets only.

(d) Finally, it should be stated if and how the presented modeling approach is reliable with respect to simulations of other wind turbines and different wind conditions. Which part of the modeling still comprises uncertainties? What would be suggestions for further developments on the proposed modeling?

**3.   Technical corrections**

- p.1: Abstract: state in one sentence which turbulence model performed best under which flow conditions.

- p.1, l.17: "... CFD code and to the..."

- p.1, l.19: "... large wind turbines...", specify what "large" and "small scale" (l.21) is. $D = 10m$ are still model scale

- p.2, l.25: "... design conditions." Specify what these design conditions are (TSR=?)

- p.3, l.21: "... created by a bi-planar..."

- p.5, l.28: "... a triangular and a trapezoidal distribution." (word trapeze/trapezoidal reoccurs at several places in text an tables).

- p.6, Table 2: as mentioned above: a plot showing the different distributions would be illustrative.

- p.7, Table 4: as mentioned above: what does "undistributed" mean?

- p.7, l.1: " ... ACDs are?"

- p.7, l.4: "... thrust coefficients $C_T = (...)$ wind turbine are..."

- p.10, Fig.5 (a) and (b), p.11, l.5 and p.13, l.4: it is first stated that the k-epsilon and the KL k-epsilon model produce similar results on p.11, while on p.13 it is stated that the RNG k-epsilon tends to underpredict the wake recovery. Judging from Figures 5 (a) and (b) I hardly see any difference in the results by the k-epsilon and RNG k-epsilon model.

- p.13, l.1: it is stated that the TKE profiles in Fig. 5(b) are "not successfully predicted by any of the turbulence models". Could you elaborate on reasons for this giving a source from literature? Is this due to the weak performance of RANS in low turbulent environments? Is there an influence of the non-existence of tip-vortex-shedding in ACD models on the TKE profiles?

- p.14, l.1: "...capture the position of the tip vortex apart from the polynomial." Why are there several peaks appearing in Fig. 6(b)? Can you double-check for convergence?

- p.16, l.22-25: "As the purpose (...) from representing differently (...) using different turbulence models." This is a very long and hard-to-understand sentence; especially the "representing differently" part needs revision.

---

## Referee Comment (RC1) · P. van der Laan (Referee) · 9 May 2017

**Review of **Validation of the actuator disc approach using small scale model wind turbines** by Nikolaos Simisiroglou et al.**

Reviewer: M. Paul van der Laan, DTU Wind Energy

May 9, 2017

The article presents a verification and a validation of an actuator disk model using a grid refinement study and measurements of small scale wind turbines in several configurations, respectively. The influence of different force distributions and turbulence models are discussed.

**Main comments**

You have chosen to separate the presentation of the results from the discussion of the results. I think it makes more sense to combine the results section with the discussion section and call it Results and Discussion. Then you can introduce a figure or table and directly discuss it, which is more natural to read. You could also make subsections in this section, e.g. a subsection about the grid study, a subsection about the validation with measurements, etc.

The proposed Actuator Disk (AD) method could be described in more detail, see specific comments below.

The results of the grid study show that your AD method does not converge monotonically for the investigated grid sizes. I think it is important to convince the reader that your AD method converges monotonically with grid size and I suggest that you redo this study based on the specific comments below.

**Specific comments**

1. Page 1, Abstract: In the abstract you have mentioned that the method has a low computational effort; however, the article lacks the results to show this. In addition, you have mentioned that a grid spacing of 40 cells per rotor diameter is sufficient, but this means that you need quite a lot of cells for wind farm simulations. Other RANS AD modelers would use 8 or 10 cells over a rotor diameter in wind farm simulations, where the grid spacing is based on a grid study.

2. Page 2, line 28: It would be worth to mention that a constant pressure drop resembles a uniformly distributed thrust force. Do you have any idea why this was chosen in the model wind turbine design? Real modern wind turbines have non-uniform thrust distributions.

3. Page 3, line 1: You mention that the blockage ratio including the tower is 12%. If I calculate this ratio (without the tower) I get: $0.447^2\pi/(2.701h(x = 3.66)) = 12.8\%$, where I have assumed that the height $h$ of the wind tunnel is increasing linearly with downstream distance $x$: $h(x) = 0.05/11.150x + 1.801$; $h(x = 3.66) = 1.817$ m. Maybe I misunderstood something?

4. Section 2.1: You could mention the Reynolds number of the experiments.

5. Pages 5-6, AD method:

(a) You use a reference to define your AD method, but I think it would be useful to present the complete method here since the article is focused on the validation of your AD method.

(b) Eq. 8: How is the axial induction factor $\alpha_i$ calculated? Do you use $\alpha_i = \frac{1}{2}(1 - \sqrt{1 - C_T})$, implying that $\alpha_i$ is always a constant? If that is the case I would remove the index $i$ and just write $\alpha$.

(c) If you use $\alpha_i = \frac{1}{2}(1 - \sqrt{1 - C_T})$, are you aware that this relation can result in an overpredicted thrust and power? See for example the results of the AD induction method in van der Laan et al. [2]. How do you solve this issue in your AD method?

(d) If $\alpha_i$ is a constant and the inflow is uniform then the undistributed thrust distribution would always be the same as the uniformly distributed thrust, or do I miss something here? In other words, what is the difference in force distribution between Cases A and B in Table 4?

(e) Table 2: The Trapeze distribution is not defined for $0 < r < 0.2R$, what do you use in this region? It would be helpful to plot the different distributions.

6. Page 7-8, grid study:

(a) Figure 2 shows that the wake solution is highly grid dependent. Based on these results it is difficult to state which grid is fine enough. You probably need to run a finer grid level to show grid convergence. This is also confirmed by the percentage of oscillating convergence at $x = 10R$ (45%) as listed Table 5. It probably means that your results are not yet in the range where the solution is monotonically converging with grid size.

(b) Table 5, $\bar{p}$: From the grid study you have calculated an average order of accuracy of 3.58 and 5.92 at $x = 2R$ and $x = 10R$, respectively. How do you explain that the order of accuracy is so high considering the fact that you use a second order numerical scheme (central difference scheme)? You could try to also include a fourth grid level and perform a mixed order analysis as used in Réthoré et al. [1] or in van der Laan et al. (2015).

(c) Currently the grid study only covers one doubling of the coarsest grid. It would be more useful to look at bigger range of grid sizes, e.g. 5, 10, 20 and 40 or 10, 20, 40 and 80 cells over a rotor diameter.

(d) How is the total thrust force behaving with grid size? If the total thrust force is oscillating with grid size it might be the reason why the results of the grid study are unsatisfactory.

(e) Which turbulence model is used in the grid study? You could choose to perform a more basic grid study of your AD method by simulating an AD in a laminar flow (e.g. Re=100) without wind tunnel wall, such that you do not need a turbulence model.

7. Page 10, Figure 4: You could normalize the velocity contours with the freestream velocity at hub height.

8. Page 11, Figure 6: Is the thrust coefficient different for each thrust force distribution (as shown for the double wind turbines cases in Table 8)? If this is the case, it is difficult to isolate the influence of the thrust distribution on the wake flow. Ideally one could keep the the total thrust force constant and only change the distribution.

9. Page 13, Lines 1-2: Based on Figure 5b, it does not seem that the $k$-$\varepsilon$ model predicts similar $k$ levels compared to the measurements, since it produces 2-7 times smaller $k$ levels at $r = \pm R$.

10. Page 16, Lines 5-10 and Table 6. I think a 10% difference in thrust coefficient is quite a lot, especially for the upstream wind turbine.

11. Page 17, Lines 4-5: Maybe you should note that the uniformly distributed thrust might not be the best one if a real size wind turbine is modeled that typically has a non-uniform thrust distribution.

**References**

[1] Réthoré, P.-E., van der Laan, M. P., Troldborg, N., Zahle, F., and Sørensen, N. N. Verification and validation of an actuator disc model. *Wind Energy*, 17:919–937, 2014.

[2] van der Laan, M. P., Sørensen, N. N., Réthoré, P.-E., Mann, J., Kelly, M. C., and Troldborg, N. The $k$-$\varepsilon$-$f_p$ model applied to double wind turbine wakes using different actuator disk force methods. *Wind Energy*, 18(12):2223–2240, December 2015.

---

## Author Comment (AC1) · 12 Jul 2017

Dear Referee,

We would like to truly thank you for putting so much time and effort to provide for a thorough and detailed review of our manuscript.

Please find attached a copy of our revised manuscript and a point-to-point response to the reviewer's comments in the form of a supplement.

We hope that we have now addressed all comments raised and the manuscript has reached adequate quality to warrant publication.

Yours sincerely,

[Figure]

Nikolaos Simisiroglou

Please also note the supplement to this comment:
https://www.wind-energ-sci-discuss.net/wes-2017-18/wes-2017-18-AC1-
supplement.zip

―――――――――――――――――――

---

## Author Comment (AC2) · 12 Jul 2017

Dear Reviewer,

We would like to truly thank you for putting so much time and effort to provide for a thorough and detailed review of our manuscript.

Please find attached a copy of our revised manuscript and a point-to-point response to the Reviewers comments in the form of a supplement.

Yours sincerely,

Nikolaos Simisiroglou

[Figure]

Please also note the supplement to this comment: https://www.wind-energ-sci-discuss.net/wes-2017-18/wes-2017-18-AC2-supplement.zip

———————————————————

---

## Author Response (AR1)

[revised manuscript text omitted]